# Backward Chaining Method for Teaching Long-Term Care Residents to Stand Up from the Floor: A Pilot Randomized Controlled Trial

**DOI:** 10.3390/jcm14155293

**Published:** 2025-07-26

**Authors:** Anna Zsófia Kubik, Zsigmond Gyombolai, András Simon, Éva Kovács

**Affiliations:** 1Doctoral College, Semmelweis University, Health Sciences Division, 1085 Budapest, Hungary; kubik.anna.zsofia@gmail.com (A.Z.K.); gyombolai.zsigmond@semmelweis.hu (Z.G.); 2Faculty of Health Sciences, Department of Physiotherapy, Semmelweis University, 1085 Budapest, Hungary; simon.andras@semmelweis.hu; 3Faculty of Health Sciences, Department of Morphology and Physiology, Semmelweis University, 1085 Budapest, Hungary

**Keywords:** backward chaining method, floor-transfer, fear of falling, life-space mobility, older adults, long-term care

## Abstract

**Objectives**: Older adults who worry about not being able to stand up from the floor after a fall, reduce their physical activity, which leads to a higher risk of falling. The Backward Chaining Method (BCM) was developed specifically for this population to safely teach and practice the movement sequence required to stand up from the floor. Our aim is to evaluate the effectiveness of using the BCM to teach older adults how to stand up from the floor, and to determine whether this training has an impact on functional mobility, muscle strength, fear of falling, and life-space mobility. **Methods**: A total of 26 residents of a long-term care facility were randomly allocated to two groups. Residents in the intervention group (IG, *n* = 13) participated in a seven-week training program to learn how to stand up from the floor with BCM, in addition to the usual care generally offered in long-term care facilities. The participants in the control group (CG, *n* = 13) received the usual care alone. The primary outcome measure was functional mobility, assessed by the Timed Up and Go test. Secondary outcome measures included functional lower limb strength, grip strength, fear of falling, and life-space mobility. The outcomes were measured at baseline and after the seven-week intervention period. **Results**: We found no significant between-group differences in functional mobility, lower limb strength and grip strength; however, IG subjects demonstrated significantly lower fear of falling scores, and significantly higher life-space mobility and independent life-space mobility scores compared to CG subjects after the training program. **Conclusions**: This study demonstrates that the Backward Chaining Method is a feasible, well-tolerated intervention in a long-term care setting and it may have meaningful benefits, particularly in lessening fear of falling and improving life-space mobility and independent life-space mobility when incorporated into the usual physiotherapy interventions.

## 1. Introduction

Getting down to and rising from the floor is a common and familiar movement in childhood as well as in play and sport activities during young adulthood. However, in middle age, daily routines offer fewer opportunities to engage in this once common movement [1]. Thus, this movement, which is very easy to perform in the first half of life, becomes more difficult as we get older [2]. Fear of being unable to stand up after a fall leads older adults to avoid potentially dangerous situations, limit their physical activities, and, over time, suffer from a restricted life space, a diminished functional capacity and ultimately, a heightened risk of falling [2,3,4,5,6]. According to national demographic data, the proportion of adults aged 65 years and older is steadily increasing in Hungary, in line with global aging trends. This shift highlights the growing relevance of interventions that support healthy aging and functional independence in older adults [7].

Teaching how to stand up from the floor is already part of the latest fall prevention guidelines [8]. However, practicing the movement of rising from the floor among older adults presents significant challenges for both the patient and the therapist. This may be due not only to actual muscular weakness, but also to the perception held either by the patient or the therapist that the elderly individual is no longer capable of assuming the starting position of the movement, namely, lying on the floor [9].

The Backward Chaining Method (BCM) was developed specifically for the elderly population to safely teach and practice the movement sequence required to stand up from the floor. The core principle of this structured, standardized teaching method is that the sequence of movements is introduced beginning with the most stable position, namely, standing, and then systematically progressing to kneeling, prone kneeling, and ultimately, lying on the floor. Progression to the next step occurs only when the older person can confidently and securely perform the preceding step. This teaching method does not require the older individual to assume a position that is considered unsafe, such as lying on the floor, at the initial stage of learning. Beyond minimizing stress, this approach reduces patients’ fear of falling and fosters greater self-confidence [2,9]. This strategy also serves as a therapeutic exercise that enhances muscular strength and improves the functional mobility of older adults [10,11]. Therefore, practicing the movement remains beneficial even if the patient is eventually unable to descend to or rise from the floor [2,9,12].

To date, a substantial body of literature exists on the effects of interventions aimed at reducing fall risk [13,14,15,16,17]. Despite extensive research on fall prevention, studies examining the effectiveness of interventions in regaining or restoring the ability to rise from the floor remain limited [18]. Moreover, an even smaller number of intervention studies have been conducted using a randomized controlled design [10,19].

Zak et al. reported a significant improvement in functional mobility measured by the Timed Up and Go test in older adults who participated in a physical rehabilitation regimen incorporating the BCM for 12 weeks, three times a week, compared to those, whose physical rehabilitation regimen was less intense, and instead of the BCM, used the conventional method in teaching standing up from the floor [10].

Hofmeyer et al. found that individuals who practiced standing up from the floor just three times a week for 45 min sessions over a two-week period demonstrated significantly greater ease and speed in performing the movement compared to those who engaged only in the seated exercise [19].

Despite the significant impact that fear of falling, reduced physical activity, and constrained life space have on the older population in long-term care, the effects of this particular teaching method within this population remain unexplored [20,21,22].

Furthermore, evidence suggests that a reduction in life space is associated with a decline in physical capabilities, an elevated risk of future falls, diminished quality of life, and an increased mortality within one year [20,21,22,23]. If older adults in long-term care were confident in their ability to rise from the floor after a fall, they might be less inclined to restrict their physical activity. Maintaining mobility could help preserve their physical function, thereby reducing the risk of falls, disability, and premature mortality.

To date, no randomized controlled trials in the literature have examined the effects of BCM on physical abilities, fear of falling, and life-space mobility in institutionalized older people.

To address this research gap, our study had two primary objectives. First, we aimed to evaluate the effects of teaching the movement sequence of rising from the floor using the BCM on functional mobility, muscle strength, fear of falling, and life-space mobility among long-term care residents. Second, we sought to assess the acceptability of this intervention within this population.

## 2. Materials and Methods

### 2.1. Study Design

This single-blinded, double-arm randomized controlled trial was conducted among long-term care residents in Budapest, Hungary, in which all residents who met the eligibility criteria participated. While both the participants and physiotherapists administering the BCM were unblinded, the assessors remained blinded. The study was designed and conducted in accordance with the Declaration of Helsinki and reported following the Consolidated Standards of Reporting Trials (CONSORT) guidelines. After approval was obtained from the Institutional Review Board, the study was registered on ClinicalTrials.gov under the identifier NCT06908317.

### 2.2. Participants and Randomization

Residents were included in the study if they were 65 years of age or above, had resided in the institution for at least two months, and were able to walk independently, or with the aid of an assistive device. Residents were excluded from the study if they were unfit to be taught how to rise from the floor due to severe pain (Visual Analog Scale > 7/10), physical limitations, lower limb endoprosthesis surgery within the preceding six months, cognitive impairment preventing co-operation (as determined by psychic assessment), in the case of high blood pressure (over 160/90 mmHg) or in the case of a planned relocation from the institution during the study period. Figure 1 presents the enrollment process of the study.

Eligible individuals were informed about the aim and process of the study, as well as their right to withdraw at any stage of the trial. Informed consent was then obtained from all participants. Following baseline assessment, eligible participants were randomized using stratified randomization, accounting for walking aid usage and baseline FES-I scores to either the intervention group (IG) or the control group (CG). The random allocation sequence was generated using a computer-based method (GraphPad) by an independent researcher who was blinded to both the intervention and assessment, ensuring allocation concealment.

### 2.3. Control Group Activity

Participants in the control group received usual care, which consisted of a 40 min-long chair-based, low-intensity exercise program combined with communicative social activities delivered as routinely provided in long-term care facilities. Participants were given the opportunity to participate in a subsequent instructional period focusing on floor-to-standing transitions after the conclusion of the trial period.

### 2.4. Interventional Group Activity

Participants in the intervention group, in addition to receiving the usual care generally offered in long-term care facilities, were engaged in practicing floor recovery techniques taught using the Backward Chaining Method (BCM). Sessions were conducted three times a week over a seven-week period. Each 40 min session began with a five-minute warm-up incorporating flexibility exercises and ended with a five-minute cool-down consisting of stretching and breathing exercises. The warm-up was followed by structured instruction and practice of floor-to-stand transitions utilizing the BCM. The BCM based on Reece and Simpson [24] entails practicing a structured sequence of movements designed to facilitate a safe and efficient transition from the ground to a standing position. This process is divided into seven distinct steps, as illustrated in Figure 2 and presented in Table 1.

The BCM sessions were conducted in a designated community room within the institute and were led by a physiotherapist with extensive expertise in geriatric physiotherapy. For safety and support during the exercises, additional supervision by two physiotherapy students and one caregiver was provided. In addition, blood pressure was measured prior to each session. Participants with systolic blood pressure exceeding 160 mmHg or diastolic blood pressure above 90 mmHg were excluded from that session. The practice sessions required minimal equipment, including stable chairs, wedge pillows, and training mats.

### 2.5. Measures

Participants’ age and health-related characteristics, including weight, height, Mini Mental State Examination (MMSE) scores, use of walking aids, incidence of falls within the preceding six months, and presence of chronic diseases were obtained from nursing records. Outcome assessments were conducted at baseline and after the intervention period by a trained physiotherapist. The physiotherapist was blinded to group allocation and was not involved in the administration of the exercise programs. Participants were instructed not to disclose details of their treatment to the outcome assessor.

The primary outcome measure was functional mobility, assessed using the Timed Up and Go (TUG) test, which is a widely used measurement in the Hungarian geriatric practice [25]. This test quantifies the time (in seconds) required for a participant to rise from a standard armchair with a seat height of approximately 46 cm and arm height of 65 cm, walk three meters to a cone, turn around, return to the chair, and sit down again. Participants were instructed to walk at a comfortable and safe pace. They were permitted to wear their usual footwear, use their usual walking aids, and use the chair arms for support when standing up. Physical assistance was not provided. After an initial familiarization trial, two successive test performances were recorded, and the mean value was used for analyses. If needed, a 30 s rest period was allowed between trials [26].

Secondary outcome measures included functional lower limb strength, grip strength, fear of falling and the extent and frequency of mobility, commonly referred to as life-space mobility.

Functional lower limb muscle strength was assessed using the 30 s Sit-to-Stand Test, which measures the number of sit-to-stand repetitions completed within 30 s. Participants performed the test from a chair with an approximate seat height of 46 cm, with their arms folded across the chest. They were instructed to “stand-up completely and sit down as quickly as possible”. The digital stopwatch was started on the command “go”. Before timing, each participant was asked to perform a single sit-to-stand trial. During the test, the assessor silently counted each correctly executed stand to ensure accurate performance feedback. Incorrectly performed repetitions were excluded from the final count. At the end of the 30 s test period, any stand completed more than halfway up was counted as a full repetition. Participants unable to perform a single sit to stand using a standard technique, received a score of zero. During testing, all participants wore their regular footwear [27].

Grip strength, which has been shown to strongly correlate with overall muscle strength, was assessed using the Kern MAP 40K1 dynamometer and measured in kilograms. This test was performed while seated with the dominant hand assessed by averaging two consecutive measurements [28].

Fear of falling was measured using the Hungarian version of short FES-I, which consists of seven items assessing an older adult’s level of concern about falling during routine daily activities. Responses are recorded on a four-point scale: 1 = not at all concerned, 2 = somewhat concerned, 3 = fairly concerned, 4 = very concerned. The total score ranges from 7 to 28 points, with higher scores indicating greater concern about falling. The Hungarian version of Falls Efficacy Scale International is a valid and reliable tool for measuring fear of falling among Hungarian-speaking older adults and is widely used in clinical practice and scientific studies [29].

Life-space mobility was assessed using the Nursing Home Life-Space Diameter (NHLSD). As part of the standard documentation of the long term-care facility, this measure was completed with the assistance of the staff, based on the resident’s movement patterns over the preceding two weeks. The NHLSD evaluates the extent of the resident’s mobility across four domains: movement within the (1) resident’s room, (2) within the unit, (3) outside the unit, and (4) beyond the facility. Additionally, it measures movement frequency using a six-point scale: 0 = never, 1 = less than weekly, 2 = at least weekly, 3 = more than twice per week, 4 = one to three times per day, and 5 = more than three times per day. The total NHLSD score is calculated using the formula: 1 (diameter 1 × frequency 1) + 2 (diameter 2 × frequency 2) + 3 (diameter 3 × frequency 3) + 4 (diameter 4 × frequency 4). In addition to measuring distance and frequency, the NHLSD assesses the level of human assistance required for mobility: if mobility was performed without human assistance, the score for each item was multiplied by two. Ultimately, two types of total scores were computed. One of the total scores represents the spatial range of a person’s movement, with values ranging from 0 to 50, where 0 signifies being bedbound and 50 indicates daily movement outside the facility [30]. The second total score integrates mobility independence into the overall calculation, thereby accounting not only for the distance and frequency of movement, but also for the extent of dependence on human assistance. In this scoring system, the maximum possible value, NHLSD-dependence, is 100 points [31].

### 2.6. Acceptability

Acceptability of the program was assessed based on participants’ adherence rate, and the occurrence of adverse events. The adherence was quantified by the number of completed sessions. Sufficient adherence for inclusion in the per protocol analysis was determined at 80%. To promote adherence, the significance of the study was clearly articulated, and the safety of the exercise sessions was emphasized. Participants received reminders for upcoming sessions, and individual progress in movement execution was emphasized. They were encouraged to share their positive or possible negative experiences with peers or staff members. Attendance, progress, and experiences were documented in a pre-designed attendance-sheet maintained by the physiotherapist leading the sessions.

Adverse events, including falls, were recorded throughout the intervention period by the long-term care facility staff. Staff members were provided with a pre-designed diary to record daily fall incidents. These diaries were collected weekly by the assessor physiotherapists [14].

A fall was defined as an “unexpected event in which a participant comes to rest on the ground, floor, or a lower level” [8]. A fall may include incidents, such as falling backward onto a bed or chair while attempting to stand from a seated position or losing balance and falling against a wall or piece of furniture.

### 2.7. Sample Size Estimation

Zak et al. reported a 5.3 s improvement in functional mobility, as measured using the TUG test, among older people following a teaching program utilizing the Backward Chaining Method [10]. This difference is considered clinically significant and is particularly relevant for elderly individuals with knee osteoarthritis [32], with Parkinson’s disease [33,34], Alzheimer’s disease [35], or stroke [36]. To detect this between-group difference, with a significance level of alpha = 0.05 and a statistical power of 80%, while accounting for a 30% drop-out rate, a minimum of 13 participants per group was required. The sample size calculation was performed using an online calculator available at OpenEpi (https://www.openepi.com/SampleSize/SSMean.htm, accessed on24 July 2024).

### 2.8. Data Analyses

Data were reported as numbers and percentages for discrete variables and as means with standard deviations (SDs) or median and interquartile ranges for continuous variables, depending on the distribution. Baseline comparisons between the intervention and control groups were conducted using chi-square tests for discrete variables, while independent-sample *t*-tests or Mann–Whitney U tests were applied for continuous variables, as appropriate. The Kolmogorov–Smirnov test was used to check the distribution of our data.

For each outcome variable, the analyses followed an intention to treat approach, in which all participants were analyzed within their originally (and randomly) assigned groups, followed by a per-protocol analysis, in which only those who adhered to their assigned therapeutic plan were included.

All assumptions of analysis of covariance (ANCOVA), including linearity, homogeneity of regression slope, homoscedasticity and homogeneity of variance (Levene’s test) were evaluated prior to all analyses. ANCOVA models were employed to compare post-intervention outcome variables for grip strength, the Falls Efficacy Scale International (FES-I), the Nursing Home Life-Space Diameter (NHLSD), and independent NHLSD scores with baseline scores of the dependent variable used as covariates. For the Timed Up and Go (TUG) test, and the 30 s Sit-to-Stand Test, between-group comparisons at post-intervention were conducted using Mann–Whitney U tests. In cases where significant differences were identified, effect sizes were calculated using the eta-square (η^2^), with values interpreted as follows: 0.010–0.059 representing a small effect, 0.060–0.139 a medium, and values more than 0.140 indicating a large effect.

To examine within-group changes across the intervention period, paired samples *t*-tests or Wilcoxon signed-rank tests were applied.

Cohen’s d was used to quantify within-group effect sizes, with values of 0.8 considered large, 0.5 moderate, and 0.2 small [37]. Statistical analyses were performed using SPSS 18.0 software, with statistical significance set at *p* < 0.05.

## 3. Results

After the initial screening of the residents, 26 participants were randomly assigned to the intervention group (IG, *n* = 13) or the control group (CG, *n* = 13). The progression of the selection of the participants is illustrated in a flowchart in Figure 1. A total of 24 participants (92%) adhered to the study protocol. During the trial period, two participants experienced falls, none of which resulted in injuries; therefore, both continued to participate in the program. No other adverse events were reported during the study period.

### 3.1. Baseline Characteristics

Table 2 presents the demographic and clinical characteristics of the study participants. There were no significant differences between groups across any baseline characteristics. Among the participants, 11 individuals were able to walk without any walking aids (IG: *n* = 6, 46%; CG: *n* = 5, 38.5%). One participant used a stick (CG: *n* = 1, 76.9). Four participants required a walking frame (IG: *n* = 2, 15.4%; CG: *n* = 2, 15.4%), while 10 individuals used a rollator for mobility (IG: *n* = 5, 38.5%; CG: *n* = 5, 38.5%).

### 3.2. Outcomes

The outcome data are presented in Table 3.

#### 3.2.1. Functional Mobility

No significant difference in TUG test performance was detected between groups at the end of the seven-week intervention period (z = −1.410, *p* = 0.158). Wilcoxon signed-rank tests indicated that there was no significant change in TUG test results within the intervention group between baseline and post-intervention (z = −0.471, *p* = 0.638). Similarly, no significant difference was found in the control group’s TUG test scores between baseline and post-intervention (z = −0.089, *p* = 0.929).

#### 3.2.2. Functional Lower Limb Muscle Strength

No significant difference in 30 s Sit-to-Stand Test performance was identified between groups following the seven-week intervention (z = −1.291, *p* = 0.197). Wilcoxon signed-rank tests indicated no significant change in the intervention group’s 30 s Sit-to-Stand Test scores between baseline and post-intervention (z = −1.472, *p* = 0.141). Similarly, no significant difference was found within the control group between baseline and post-intervention (z = −0.135, *p* = 0.893).

#### 3.2.3. Grip Strength

At the end of the seven-week intervention, no significant difference in grip strength was observed between groups F (1, 23) = 0.265, *p* = 0.612. Neither the intervention group (t (12) = −0.239, *p* = 0.815) nor the control group (t (12) = 256, *p* = 0.802) demonstrated any change in grip strength.

#### 3.2.4. Fear of Falling

A significant difference in FES-I scores was found between groups after adjusting for baseline values, F (1, 23) = 13.16, *p* = 0.001 with a large effect size (η^2^ = 0.364). Paired *t*-tests indicated a significant improvement in FES-I scores within the intervention group from baseline to post-intervention (t (12) = 3.371, *p* = 0.006, Cohen’s d = 0.63, indicating a medium effect). However, no significant change was observed in the control group (t (12) = −1.822, *p* = 0.093). Results are illustrated in Figure 3 (A).

#### 3.2.5. Life-Space Mobility

A significant difference in life-space mobility was found between groups after adjusting for baseline values (F (1, 23) = 10.46, *p* = 0.004), with a large effect size (η^2^ = 0.313). Paired *t*-tests indicated a significant improvement in life-space mobility within the intervention group from baseline to post-intervention (t (12) = −2.441, *p* = 0.031, Cohen’s d = 0.44 indicating a medium effect). No significant effect was observed in the control group (t (12) = 1.94, *p* = 0.076). Results are illustrated in Figure 3 (B).

#### 3.2.6. Independent Life-Space Mobility

A significant difference in independent life-space mobility was found between groups after adjusting for baseline values, (F (1, 23) = 6.975, *p* = 0.015), with a large effect size (η^2^ = 0.233). Paired *t*-tests indicated a significant improvement in independent life-space mobility within the intervention group between baseline and post-intervention (t(12) = −2.757, *p* = 0.017, Cohen’s d = 0.24, representing a small effect). However, no significant change was found in the control group (t (12) = −1.935, *p* = 0.077). Results are illustrated in Figure 3 (C).

## 4. Discussion

This randomized, controlled, assessor-blinded study was conducted to evaluate both the acceptability and effectiveness of teaching floor-to-standing transitions using the Backward Chaining Method. The study examined its impact on functional mobility, muscle strength, fear of falling, and life-space mobility among long-term care residents.

Our data suggest that BCM is an acceptable intervention among long-term care residents: the intervention group (IG) showed a mean attendance rate of 94%, with four participants attending all 20 training sessions, one participant missing three sessions, and the rest of the participants missing only one or two sessions.

We found no significant between-group differences in functional mobility and lower limb strength; however, our findings indicate that IG subjects, after participating in the BCM training program, demonstrated significantly lower fear of falling scores, and significantly higher life-space mobility and independent life-space mobility scores compared to those in the control group (CG) who received only usual care.

The lack of significant improvement in functional mobility is in contrast to the results of Zak et al. who conducted a 12-week rehabilitation regime incorporating BCM three times per week among community-dwelling older adults above 80 years of age. They found a significant improvement in TUG time, compared to their control group, which received a less intensive rehabilitation regime and learned floor transfer using the conventional method. These results suggest that not only the duration and frequency of training, but also the choice of floor-transfer strategy and the functional status of the population, may influence functional mobility outcomes [10].

Fear of falling measured by the Falls Efficacy Scale International (FES-I) is an established predictor of falls [38]. Therefore, interventions that successfully reduce FES-I scores may have benefits for fall prevention, self-efficacy, and overall physical activity. Several previous studies explored various approaches to reduce fear of falling. In a randomized controlled trial, Gusi et al. conducted a 12-week balance training program in institutionalized older adults with fear of falling. After completion of the training program, they reported a significant improvement in FES-I scores. Although the design and goal of the intervention differed from ours, the examined population, the outcome measure and the overall training load allow for meaningful comparison. Our findings align with theirs, that fear of falling can be improved by a structured exercise program in long-term care residents [13].

The measurement of life-space mobility is much more variable than the measurement of fear of falling. As seen in a recent systematic scoping review by Seinsche et al., studies examining life-space mobility are varying in outcome measures, examined populations, and intervention methods, as well. According to their review, to quantify life space, the majority of studies use the Life-Space Assessment (LSA), whereas some use the Life-Space Questionnaire, GPS or accelerometer-based measurements, or, as in our study, the NHLSD [39].

In an eight-week group-based physical and cognitive intervention conducted by Tanaka et al., the authors found no significant improvement in life-space mobility measured by the NHLSD among long-term care residents with dementia. Although the length of the intervention was similar to ours, frequency was two sessions per week, resulting in fewer total training sessions. This difference in the training load, or the different type of the intervention could be the reason behind the lack of significant improvement in NHLSD, in contrast to our own results [40].

In another study, Todo et al. conducted a three-month long, multicomponent, home-based rehabilitation program in older adults with restricted life-space mobility and reported significant improvement in LSA scores. Although differences in outcome measure and study population make comparison difficult, their findings support our conclusion that life-space mobility can be improved with structured exercise [41].

To our knowledge, this is the first study to examine the effectiveness of BCM on functional mobility, fear of falling, life-space mobility, grip strength, and lower limb strength in an institutionalized population. Although more recent guidelines incorporate teaching floor-transfer techniques into fall-prevention training programs [8], still, few long-term care facilities incorporate it into their day-to-day treatment plans. This may be in part because of the lack of knowledge, as well as due to variations in facility structures, the heavy workload pressure, and differences in standard treatment protocols. A survey by Sterke et al. found a strong agreement among physiotherapists regarding their role in fall prevention, strength and mobility training, pain management, and transfer education in nursing homes. However, they found no consensus regarding the actual interventions needed to address these areas effectively [42].

Another aspect could be that physiotherapy interventions are often examined in light of outcome measures that are irrelevant in a long-term care setting. This misalignment can hinder the integration of such interventions into routine clinical practice by facility management and healthcare professionals [43].

By using the NHLSD and implementing the BCM intervention directly within a nursing home setting, our study offers a model tailored to the realities of long-term care practice. Our findings suggest that even in institutionalized elderly care, BCM may be a practical and effective addition to usual care, as it seems to improve life-space mobility and lessen fear of falling.

Our study has several limitations that may explain the lack of statistically significant results in functional mobility and muscle strength. Firstly, the small sample size, consisting only of women, limits the generalizability of our findings. Secondly, the seven-week intervention period may have been insufficient to stimulate measurable improvements in the outcome measures. Although some previous studies on the BCM examined even shorter training periods [19,44], most effective interventions typically last 12 weeks or longer [10,13,41]. Furthermore, participants began the program with relatively good functional status, which may have limited the potential for noticeable improvement. These factors together likely contributed to the non-significant results in functional measures. Finally, no qualitative data were collected. In retrospect, this was a missed opportunity to explore the participants’ experiences and perceptions regarding the training program and its effect. While we did receive informal verbal feedback from some participants, caregivers and healthcare professionals, a more systematic approach, such as the collection of qualitative data via semi-structured interviews, could have provided deeper insight into the subjective impact of the intervention. Additionally, collecting quantitative data on mental health could open further areas for research.

## 5. Conclusions

This study demonstrates that the Backward Chaining Method is a feasible, well-tolerated intervention in a long-term care setting. Our data suggest that incorporating BCM into the usual physiotherapy interventions may have meaningful benefits, particularly in lessening fear of falling and improving life-space and independent life-space mobility. Future research should explore longer intervention periods, larger and more diverse samples, and incorporate additional measures regarding mental health, to address both psychological and physical contributors to fall risk.

## Figures and Tables

**Figure 1 jcm-14-05293-f001:**
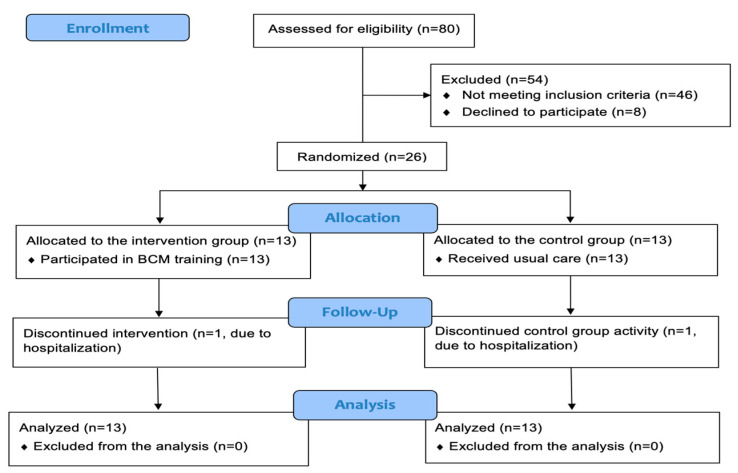
CONSORT flow diagram detailing the enrollment process of the study.

**Figure 2 jcm-14-05293-f002:**
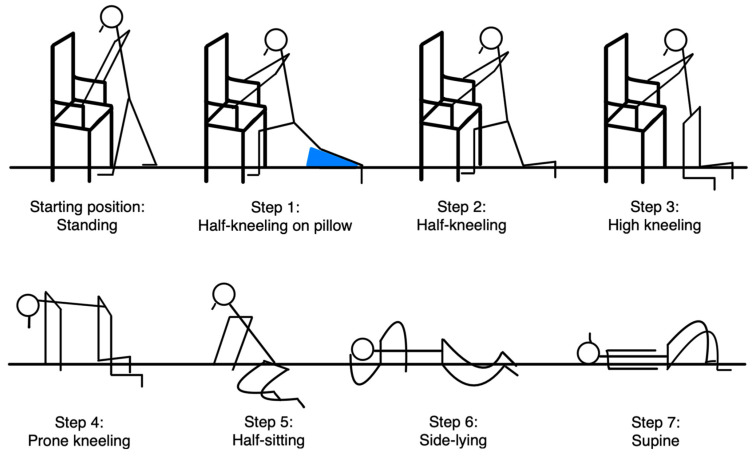
Steps of the Backward Chaining Method based on Reece and Simpson [24].

**Figure 3 jcm-14-05293-f003:**
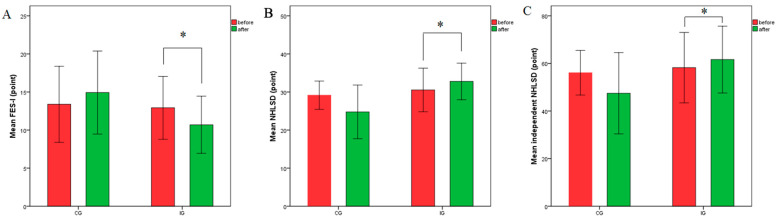
Effects of BCM on subjects’ (**A**) FES-I, (**B**) NHLSD, and (**C**) independent NHLSD. Values are mean ± SD. * *p*-value < 0.05; ** *p*-value < 0.01. Abbreviations: IG: intervention group; CG: control group; FES-I: Falls Efficacy Scale International; NHLSD: Nursing Home Life-Space Diameter.

**Table 1 jcm-14-05293-t001:** Steps of the Backward Chaining Method based on Reece and Simpson [24].

Step 1	Descending from standing one foot forward to half-kneeling on a wedge pillow. Once step 1 is completed, the participant returns to the standing position.
Step 2	Descending from the standing position to half-kneeling on the floor. Once step 2 is completed, the participant returns to the standing position.
Step 3	Following step 2, the participant lowers both knees to the floor into a high kneeling position. Once step 3 is completed, the participant returns to the standing position through step 2.
Step 4	Following step 3, the participant descends both hands to the floor one at a time, arriving to the prone kneeling position. Once step 4 is completed, the participant returns to the standing position through steps 3 and 2.
Step 5	Following step 4, the participant descends the body to half-sitting. A pillow can be positioned under the hip to soften the floor surface. Once step 5 is completed, the participant returns to the standing position through steps 4, 3 and 2.
Step 6	Following step 5, the participant descends the body to the side-lying position. Once step 6 is completed, the participant returns to the standing position through steps 5, 4, 3, and 2.
Step 7	From side-lying, the participant turns into a supine position. To return to standing, all previous steps are reversed in sequence: supine > side-lying > half-sitting > prone kneeling > high kneeling > half-kneeling > standing.

**Table 2 jcm-14-05293-t002:** Demographic and clinical characteristics of study participants.

Participants’ Characteristics	IG (*n* = 13)	CG (*n* = 13)	*p*-Value
Age (years), mean (SD)	82.7 (7.8)	87.7 (7.1)	0.110
BMI (kg/m^2^), mean (SD)	25.93 (4.3)	25.39 (3.8)	0.735
MMSE (points), mean (SD)	23.54 (5.7)	21.92 (6.4)	0.505
Barthel index (points), mean (SD)	94.62 (6.9)	91.15 (9.4)	0.295
Persons who fell in the last year, n (%)	10 (76.9)	8 (61.5)	0.395
Chronic diseases			
Cardiological disease, n (%)	3 (23.1)	4 (30.8)	0.658
Pulmonological disease, n (%)	1 (7.7)	4 (30.8)	0.320
Diabetes mellitus, n (%)	2 (15.4)	1 (7.7)	0.539
Hypertension, n (%)	9 (69)	12 (93)	0.320
Lower limb arthritis, n (%)	2 (15.4)	2 (15.4)	1.000
Osteoporosis, n (%)	4 (30.8)	3 (23.1)	0.658
Persons with hearing aids, n (%)	3 (23.1)	5 (38.5)	0.671

SD: standard deviation; BMI: body mass index; MMSE: Mini Mental State Examination.

**Table 3 jcm-14-05293-t003:** Outcome measures at baseline and after the intervention periods in groups.

	IG		CG	
	At Baseline	After seven Weeks	At Baseline	After seven Weeks
TUG ^a^ (s)	13 (10.8–30.8)	14.5 (10.8–28.1)	13 (10.8–30.8)	20.8 (16.4–30.1)
30 s STS Test ^a^ (number)	9 (4.5–10.5)	7 (2.75–10.5)	4.5 (0–9.25)	4 (0–7.5)
Grip strength ^b^ (kg)	12.65 (5.7)	12.95 (5.5)	12.1 (3.6)	11.9 (3.6)
FES-I ^b^ (point)	12.92 (4.13)	10.69 (3.75)	13.38 (4.99)	14.92 (5.45)
NHLSD ^b^ (point)	30.54 (5.7)	32.77 (4.8)	29.15 (3.7)	24.77 (7.04)
Independent NHLSD ^b^ (point)	58.23 (14.8)	61.62 (14.02)	56.08 (9.4)	47.46 (17.08)

IG: intervention group; CG: control group; TUG: Timed Up and Go; STS: Sit to Stand; FES-I: Falls Efficacy Scale International; NHLSD: Nursing Home Life-Space Diameter; ^a^ values are median (Q1–Q3), ^b^ values are mean (SD).

## Data Availability

The raw data supporting the conclusions of this article can be obtained on request from the corresponding author.

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
