# Peer review of "Backward Chaining Method for Teaching Long-Term Care Residents to Stand Up from the Floor: A Pilot Randomized Controlled Trial"

_jcm, 2025, doi:10.3390/jcm14155293_

Round 1
Reviewer 1 Report
Comments and Suggestions for Authors
Dear Authors,
First and foremost, I would like to commend you for submitting this manuscript, which addresses a highly relevant and timely issue in the field of geriatric rehabilitation: the empowerment of older adults residing in long-term care facilities to independently get up from the floor after a fall, using the Backward Chaining Method. This topic holds significant implications for functional autonomy, fall prevention, and mobility promotion, particularly in institutional settings where research remains scarce.
The manuscript presents a solid methodological design (a single-blinded randomised controlled trial), with commendable participant adherence and promising results regarding reductions in fear of falling and improvements in life-space mobility—findings that are statistically significant and clinically meaningful.
However, I recommend that the authors consider the following points:
1. The title is too long and could be more concise.
2. Structural revision of tables and figures, particularly with regard to their graphic layout and typographical organisation, which currently make them difficult to understand. Greater clarity and consistency in presentation is recommended.
3. Clarification of the inclusion criteria, explicitly defining how participant participation was quantified and what threshold was used to determine sufficient adherence for inclusion in the per protocol analysis.
4. Justification and contextualisation of the measurement tools used (FES-I, TUG, Sit-to-Stand, NHLSD), especially with regard to their applicability and validation for the target population, particularly in the context of long-term care in Hungary.
5. Revision of the descriptive results section, reducing redundancy with information already contained in the tables. The inclusion of concise and interpretative comments would improve readability and conceptual clarity.
6. Critical deepening of the discussion, particularly with regard to the absence of statistically significant results in functional mobility and muscle strength. The authors should reflect on factors such as the duration of the intervention, the small sample size and the initial functional status of the participants.
7. Consider including in future suggestions the assessment of qualitative data: the lack of systematic collection of participants' and caregivers' perceptions represents a missed opportunity. Future studies could greatly benefit from the incorporation of qualitative methods to deepen the understanding of the subjective impact of the intervention.
In summary, this is a relevant and contextually innovative study with promising practical applicability in institutional care. The research helps to address a gap identified in the literature and, once the suggested revisions are implemented, may represent a valuable contribution to evidence-based practice.
Decision: Major revision required.

It is recommended that the linguistic quality of the manuscript be improved through professional revision by a native English speaker, in order to ensure greater fluency, terminological precision, and grammatical accuracy.
Reviewer 2 Report
Comments and Suggestions for Authors
The article submitted for review addresses an important topic in the field of geriatric care. The authors presented their research in an interesting way. Methodologically, the work is developed correctly. The selection of the research group described in detail. Standardized research tools were used. The results of the study developed correctly. However, the reviewer's reservation is the timeliness of the references compiled in the article. References count 43 items, of which as many as 25 are literature from more than 10 years. Please update the references in the article. The current literature adds to the value of the article.
In my opinion, the article should be improved by adding an up-to-date literature list.
Reviewer 3 Report
Comments and Suggestions for Authors
Dear authors,
The study I have analyzed is a randomized, controlled, single-blind clinical trial. The objective was to determine the efficacy of teaching the transition from floor to standing using the backward chaining method and the degree of acceptability of this method. This is an interesting study with great application in clinical practice. The following are my concerns:
INTRODUCTION
- Authors should include some epidemiological data to support their claims.
MATERIALS AND METHODS
2.6. Acceptability:
- It is not very clear how user “acceptability” was measured. The authors should explain better how they analyzed this information.
- The exclusion criterion based on blood pressure should be included in section 2.2.
REFERENCES
- Many bibliographies are obsolete. The bibliographic citations used are more than 5 years old (76 %). The authors must update and arrange the bibliography.
- Some references are incomplete or have errors. In addition, the format of the references is not uniform. The authors should review this section.
Round 2
Reviewer 2 Report
Comments and Suggestions for Authors
I recomend the publication of the article.
Reviewer 3 Report
Comments and Suggestions for Authors
Dear authors,
Thanks for your reply. Congratulations on your work.
Best regards